# Probabilistic Curve Learning: Coulomb Repulsion and the Electrostatic Gaussian Process

**Ye Wang**
Department of Statistics
Duke University
Durham, NC, USA, 27705
eric.ye.wang@duke.edu

**David Dunson**
Department of Statistics
Duke University
Durham, NC, USA, 27705
dunson@stat.duke.edu

## Abstract

Learning of low dimensional structure in multidimensional data is a canonical problem in machine learning. One common approach is to suppose that the observed data are close to a lower-dimensional smooth manifold. There are a rich variety of manifold learning methods available, which allow mapping of data points to the manifold. However, there is a clear lack of probabilistic methods that allow learning of the manifold along with the generative distribution of the observed data. The best attempt is the Gaussian process latent variable model (GP-LVM), but identifiability issues lead to poor performance. We solve these issues by proposing a novel Coulomb repulsive process (Corp) for locations of points on the manifold, inspired by physical models of electrostatic interactions among particles. Combining this process with a GP prior for the mapping function yields a novel electrostatic GP (electroGP) process. Focusing on the simple case of a one-dimensional manifold, we develop efficient inference algorithms, and illustrate substantially improved performance in a variety of experiments including filling in missing frames in video.

## 1   Introduction

There is broad interest in learning and exploiting lower-dimensional structure in high-dimensional data. A canonical case is when the low dimensional structure corresponds to a $p$-dimensional smooth Riemannian manifold $\mathcal{M}$ embedded in the $d$-dimensional ambient space $\mathcal{Y}$ of the observed data $\boldsymbol{y}$. Assuming that the observed data are close to $\mathcal{M}$, it becomes of substantial interest to learn $\mathcal{M}$ along with the mapping $\mu$ from $\mathcal{M} \to \mathcal{Y}$. This allows better data visualization and for one to exploit the lower-dimensional structure to combat the curse of dimensionality in developing efficient machine learning algorithms for a variety of tasks.

The current literature on *manifold learning* focuses on estimating the coordinates $\boldsymbol{x} \in \mathcal{M}$ corresponding to $\boldsymbol{y}$ by optimization, finding $\boldsymbol{x}$'s on the manifold $\mathcal{M}$ that preserve distances between the corresponding $\boldsymbol{y}$'s in $\mathcal{Y}$. There are many such methods, including Isomap [1], locally-linear embedding [2] and Laplacian eigenmaps [3]. Such methods have seen broad use, but have some clear limitations relative to *probabilistic manifold learning* approaches, which allow explicit learning of $\mathcal{M}$, the mapping $\mu$ and the distribution of $\boldsymbol{y}$.

There has been some considerable focus on probabilistic models, which would seem to allow learning of $\mathcal{M}$ and $\mu$. Two notable examples are mixtures of factor analyzers (MFA) [4, 5] and Gaussian process latent variable models (GP-LVM) [6]. Bayesian GP-LVM [7] is a Bayesian formulation of GP-LVM which automatically learns the intrinsic dimension $p$ and handles missing data. Such approaches are useful in exploiting lower-dimensional structure in estimating the distribution of $\boldsymbol{y}$, but unfortunately have critical problems in terms of reliable estimation of the manifold and mapping

function. MFA is not smooth in approximating the manifold with a collage of lower dimensional hyper-planes, and hence we focus further discussion on Bayesian GP-LVM. Similar problems occur for MFA and other probabilistic manifold learning methods.

In general form for the $i$th data vector, Bayesian GP-LVM lets $\boldsymbol{y}_i = \mu(\boldsymbol{x}_i) + \boldsymbol{\epsilon}_i$, with $\mu$ assigned a Gaussian process prior, $\boldsymbol{x}_i$ generated from a pre-specified Gaussian or uniform distribution over a $p$-dimensional space, and the residual $\boldsymbol{\epsilon}_i$ drawn from a $d$-dimensional Gaussian centered on zero with diagonal or spherical covariance. While this model seems appropriate to manifold learning, identifiability problems lead to extremely poor performance in estimating $\mathcal{M}$ and $\mu$. To give an intuition for the root cause of the problem, consider the case in which $\boldsymbol{x}_i$ are drawn independently from a uniform distribution over $[0, 1]^p$. The model is so flexible that we could fit the training data $\boldsymbol{y}_i$, for $i = 1, \ldots, n$, just as well if we did not use the entire hypercube but just placed all the $\boldsymbol{x}_i$ values in a small subset of $[0, 1]^p$. The uniform prior will not discourage this tendency to not spread out the latent coordinates, which unfortunately has disasterous consequences illustrated in our experiments. The structure of the model is just too flexible, and further constraints are needed. Replacing the uniform with a standard Gaussian does not solve the problem. Constrained likelihood methods [8, 9] mitigate the issue to some extent, but do not correspond to a proper Bayesian generative model.

To make the problem more tractable, we focus on the case in which $\mathcal{M}$ is a one-dimensional smooth compact manifold. Assume $\boldsymbol{y}_i = \boldsymbol{\mu}(x_i) + \boldsymbol{\epsilon}_i$, with $\boldsymbol{\epsilon}_i$ Gaussian noise, and $\boldsymbol{\mu} : (0, 1) \mapsto \mathcal{M}$ a smooth mapping such that $\mu_j(\cdot) \in C^\infty$ for $j = 1, \ldots, d$, where $\boldsymbol{\mu}(x) = (\mu_1(x), \ldots, \mu_d(x))$. We focus on finding a good estimate of $\boldsymbol{\mu}$, and hence the manifold, via a probabilistic learning framework. We refer to this problem as probabilistic curve learning (PCL) motivated by the principal curve literature [10]. PCL differs substantially from the principal curve learning problem, which seeks to estimate a non-linear curve through the data, which may be very different from the true manifold.

Our proposed approach builds on GP-LVM; in particular, our primary innovation is to generate the latent coordinates $\boldsymbol{x}_i$ from a novel repulsive process. There is an interesting literature on repulsive point process modeling ranging from various Matern processes [11] to the determinantal point process (DPP) [12]. In our very different context, these processes lead to unnecessary complexity — computationally and otherwise — and we propose a new *Coulomb repulsive process* (Corp) motivated by Coulomb's law of electrostatic interaction between electrically charged particles. Using Corp for the latent positions has the effect of strongly favoring spread out locations on the manifold, effectively solving the identifiability problem mentioned above for the GP-LVM. We refer to the GP with Corp on the latent positions as an electrostatic GP (electroGP).

The remainder of the paper is organized as follows. The Coulomb repulsive process is proposed in § 2 and the electroGP is presented in § 3 with a comparison between electroGP and GP-LVM demonstrated via simulations. The performance is further evaluated via real world datasets in § 4. A discussion is reported in § 5.

## 2 Coulomb repulsive process

### 2.1 Formulation

**Definition 1.** *A univariate process is a Coulomb repulsive process (Corp) if and only if for every finite set of indices $t_1, \ldots, t_k$ in the index set $\mathbb{N}_+$,*

$$X_{t_1} \sim unif(0, 1),$$
$$p(X_{t_i}|X_{t_1}, \ldots, X_{t_{i-1}}) \propto \Pi_{j=1}^{i-1} \sin^{2r}\left(\pi X_{t_i} - \pi X_{t_j}\right) \mathbb{1}_{X_{t_i} \in [0,1]}, \; i > 1, \quad (1)$$

*where $r > 0$ is the repulsive parameter. The process is denoted as $X_t \sim Corp(r)$.*

The process is named by its analogy in electrostatic physics where by Coulomb law, two electrostatic positive charges will repel each other by a force proportional to the reciprocal of their square distance. Letting $d(x, y) = \sin|\pi x - \pi y|$, the above conditional probability of $X_{t_i}$ given $X_{t_j}$ is proportional to $d^{2r}(X_{t_i}, X_{t_j})$, shrinking the probability exponentially fast as two states get closer to each other. Note that the periodicity of the sine function eliminates the edges of $[0, 1]$, making the electrostatic energy field homogeneous everywhere on $[0, 1]$.

Several observations related to Kolmogorov extension theorem can be made immediately, ensuring Corp to be well defined. Firstly, the conditional density defined in (1) is positive and integrable,

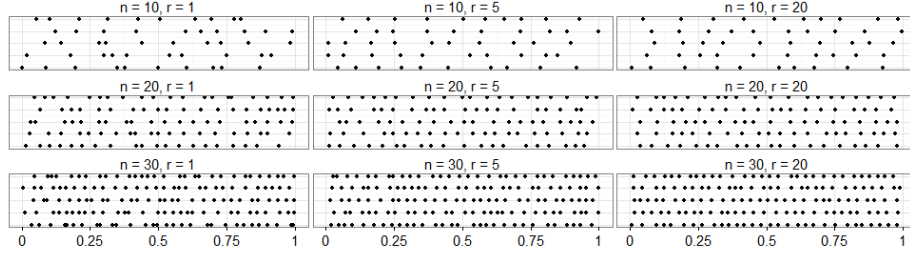

Figure 1: Each facet consists of 5 rows, with each row representing an 1-dimensional scatterplot of a random realization of Corp under certain $n$ and $r$.

since $X_t$'s are constrained in a compact interval, and $\sin^{2r}(\cdot)$ is positive and bounded. Hence, the finite distributions are well defined.

Secondly, the joint finite p.d.f. for $X_{t_1}, \ldots, X_{t_k}$ can be derived as

$$p(X_{t_1}, \ldots, X_{t_k}) \propto \Pi_{i>j} \sin^{2r}\left(\pi X_{t_i} - \pi X_{t_j}\right). \tag{2}$$

As can be easily seen, any permutation of $t_1, \ldots, t_k$ will result in the same joint finite distribution, hence this finite distribution is exchangeable.

Thirdly, it can be easily checked that for any finite set of indices $t_1, \ldots, t_{k+m}$,

$$p(X_{t_1}, \ldots, X_{t_k}) = \int_0^1 \cdots \int_0^1 p(X_{t_1}, \ldots, X_{t_k}, X_{t_{k+1}}, \ldots, X_{t_{k+m}}) \mathrm{d}X_{t_{k+1}} \ldots \mathrm{d}X_{t_{k+m}},$$

by observing that

$$p(X_{t_1}, \ldots, X_{t_k}, X_{t_{k+1}}, \ldots, X_{t_{k+m}}) = p(X_{t_1}, \ldots, X_{t_k})\Pi_{j=1}^m p(X_{t_{k+j}} | X_{t_1}, \ldots, X_{t_{k+j-1}}).$$

## 2.2 Properties

Assuming $X_t$, $t \in \mathbb{N}_+$ is a realization from Corp, then the following lemmas hold.

**Lemma 1.** *For any $n \in \mathbb{N}_+$, any $1 \leqslant i < n$ and any $\epsilon > 0$, we have*

$$p(X_n \in \mathcal{B}(X_i, \epsilon) | X_1, \ldots, X_{n-1}) < \frac{2\pi^2 \epsilon^{2r+1}}{2r+1}$$

*where $\mathcal{B}(X_i, \epsilon) = \{X \in (0, 1) : d(X, X_i) < \epsilon\}$.*

**Lemma 2.** *For any $n \in \mathbb{N}_+$, the p.d.f. (2) of $X_1, \ldots, X_n$ (due to the exchangeability, we can assume $X_1 < X_2 < \cdots < X_n$ without loss of generality) is maximized when and only when*

$$d(X_i, X_{i-1}) = \sin\left(\frac{1}{n+1}\right) \text{ for all } 2 \leqslant i \leqslant n.$$

According to Lemma 1 and Lemma 2, Corp will nudge the $x$'s to be spread out within $[0, 1]$, and penalizes the case when two $x$'s get too close. Figure 1 presents some simulations from Corp. This nudge becomes stronger as the sample size $n$ grows, or as the repulsive parameter $r$ grows. The properties of Corp makes it ideal for strongly favoring spread out latent positions across the manifold, avoiding the gaps and clustering in small regions that plague GP-LVM-type methods. The proofs for the lemmas and a simulation algorithm based on rejection sampling can be found in the supplement.

## 2.3 Multivariate Corp

**Definition 2.** *A $p$-dimensional multivariate process is a Coulomb repulsive process if and only if for every finite set of indices $t_1, \ldots, t_k$ in the index set $\mathbb{N}_+$,*

$$X_{m,t_1} \sim unif(0, 1), \text{ for } m = 1, \ldots, p$$

$$p(\boldsymbol{X}_{t_i} | \boldsymbol{X}_{t_1}, \ldots, \boldsymbol{X}_{t_{i-1}}) \propto \Pi_{j=1}^{i-1}\left[\sum_{m=1}^{p+1}(Y_{m,t_i} - Y_{m,t_j})^2\right]^r \mathbb{1}_{X_{t_i} \in (0,1)}, \ i > 1$$

*where the p-dimensional spherical coordinates $\boldsymbol{X}_t$'s have been converted into the $(p + 1)$-dimensional Cartesian coordinates $\boldsymbol{Y}_t$:*

$$Y_{1,t} = \cos(2\pi X_{1,t})$$
$$Y_{2,t} = \sin(2\pi X_{1,t})\cos(2\pi X_{2,t})$$
$$\vdots$$
$$Y_{p,t} = \sin(2\pi X_{1,t})\sin(2\pi X_{2,t})\ldots\sin(2\pi X_{p-1,t})\cos(2\pi X_{p,t})$$
$$Y_{p+1,t} = \sin(2\pi X_{1,t})\sin(2\pi X_{2,t})\ldots\sin(2\pi X_{p-1,t})\sin(2\pi X_{p,t}).$$

The multivariate Corp maps the hyper-cubic $(0, 1)^p$ through a spherical coordinate system to a unit hyper-ball in $\Re^{p+1}$. The repulsion is then defined as the reciprocal of the square Euclidean distances between these mapped points in $\Re^{p+1}$. Based on this construction of multivariate Corp, a straightfoward generalization of the electroGP model to a $p$-dimensional manifold could be made, where $p > 1$.

## 3 Electrostatic Gaussian Process

### 3.1 Formulation and Model Fitting

In this section, we propose the electrostatic Gaussian process (electroGP) model. Assuming $n$ $d$-dimensional data vectors $\boldsymbol{y}_1, \ldots, \boldsymbol{y}_n$ are observed, the model is given by

$$
\begin{aligned}
y_{i,j} &= \mu_j(x_i) + \epsilon_{i,j}, \quad \epsilon_{i,j} \sim \mathcal{N}(0, \sigma_j^2), \\
x_i &\sim \text{Corp}(r), \quad i = 1, \ldots, n, \\
\mu_j &\sim \mathcal{GP}(0, K^j), \quad j = 1, \ldots, d,
\end{aligned}
\tag{3}
$$

where $\boldsymbol{y}_i = (y_{i,1}, \ldots, y_{i,d})$ for $i = 1, \ldots, n$ and $\mathcal{GP}(0, K^j)$ denotes a Gaussian process prior with covariance function $K^j(x, y) = \phi_j \exp\{-\alpha_j(x - y)^2\}$.

Letting $\Theta = (\sigma_1^2, \alpha_1, \phi_1, \ldots, \sigma_d^2, \alpha_d, \phi_d)$ denote the model hyperparameters, model (3) could be fitted by maximizing the joint posterior distribution of $\boldsymbol{x} = (x_1, \ldots, x_n)$ and $\Theta$,

$$(\hat{\boldsymbol{x}}, \hat{\Theta}) = \arg\max_{\boldsymbol{x}, \Theta} p(\boldsymbol{x} | \boldsymbol{y}_{1:n}, \Theta, r), \tag{4}$$

where the repulsive parameter $r$ is fixed and can be tuned using cross validation. Based on our experience, setting $r = 1$ always yields good results, and hence is used as a default across this paper. For the simplicity of notations, $r$ is excluded in the remainder. The above optimization problem can be rewritten as

$$(\hat{\boldsymbol{x}}, \hat{\Theta}) = \arg\max_{\boldsymbol{x}, \Theta} \ell(\boldsymbol{y}_{1:n} | \boldsymbol{x}, \Theta) + \log\big[\pi(\boldsymbol{x})\big],$$

where $\ell(\cdot)$ denotes the log likelihood function and $\pi(\cdot)$ denotes the finite dimensional pdf of Corp. Hence the Corp prior can also be viewed as a repulsive constraint in the optimization problem.

It can be easily checked that $\log\big[\pi(x_i = x_j)\big] = -\infty$, for any $i$ and $j$. Starting at initial values $x_0$, the optimizer will converge to a local solution that maintains the same order as the initial $x_0$'s. We refer to this as the *self-truncation property*. We find that conditionally on the starting order, the optimization algorithm converges rapidly and yields stable results. Although the $x$'s are not identifiable, since the target function (4) is invariant under rotation, a unique solution does exist conditionally on the specified order.

Self-truncation raises the necessity of finding good initial values, or at least a good initial ordering for $x$'s. Fortunately, in our experience, simply applying any standard manifold learning algorithm to estimate $x_0$ in a manner that preserves distances in $\mathcal{Y}$ yields good performance. We find very similar results using LLE, Isomap and eigenmap, but focus on LLE in all our implementations. Our algorithm can be summarized as follows.

1. Learn the one dimensional coordinate $\boldsymbol{x}_0$ by your favorite distance-preserving manifold learning algorithm and rescale $\boldsymbol{x}_0$ into $(0, 1)$;

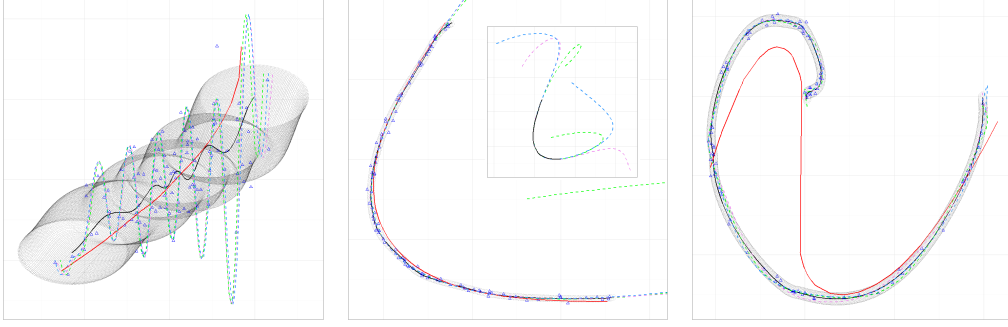

Figure 2: Visualization of three simulation experiments where the data (triangles) are simulated from a bivariate Gaussian (**left**), a rotated parabola with Gaussian noises (**middle**) and a spiral with Gaussian noises (**right**). The dotted shading denotes the 95% posterior predictive uncertainty band of $(y_1, y_2)$ under electroGP. The black curve denotes the posterior mean curve under electroGP and the red curve denotes the P-curve. The three dashed curves denote three realizations from GP-LVM. The middle panel shows a zoom-in region and the full figure is shown in the embedded box.

2. Solve $\Theta_0 = \arg\max_\Theta p(\boldsymbol{y}_{1:n}|\boldsymbol{x}_0, \Theta, r)$ using scaled conjugate gradient descent (SCG);

3. Using SCG, setting $\boldsymbol{x}_0$ and $\Theta_0$ to be the initial values, solve $\hat{\boldsymbol{x}}$ and $\hat{\Theta}$ w.r.t. (4).

## 3.2 Posterior Mean Curve and Uncertainty Bands

In this subsection, we describe how to obtain a point estimate of the curve $\boldsymbol{\mu}$ and how to characterize its uncertainty under electroGP. Such point and interval estimation is as of yet unsolved in the literature, and is of critical importance. In particular, it is difficult to interpret a single point estimate without some quantification of how uncertain that estimate is. We use the posterior mean curve $\hat{\boldsymbol{\mu}} = \mathbb{E}(\boldsymbol{\mu}|\hat{\boldsymbol{x}}, \boldsymbol{y}_{1:n}, \hat{\Theta})$ as the Bayes optimal estimator under squared error loss. As a curve, $\hat{\boldsymbol{\mu}}$ has infinite dimensions. Hence, in order to store and visualize it, we discretize $[0, 1]$ to obtain $n_\mu$ equally-spaced grid points $x_i^\mu = \frac{i-1}{n_\mu - 1}$ for $i = 1, \ldots, n_\mu$. Using basic multivariate Gaussian theory, the following expectation is easy to compute.

$$\left(\hat{\boldsymbol{\mu}}(x_1^\mu), \ldots, \hat{\boldsymbol{\mu}}(x_{n_\mu}^\mu)\right) = \mathbb{E}\left(\boldsymbol{\mu}(x_1^\mu), \ldots, \boldsymbol{\mu}(x_{n_\mu}^\mu)|\hat{\boldsymbol{x}}, \boldsymbol{y}_{1:n}, \hat{\Theta}\right).$$

Then $\hat{\boldsymbol{\mu}}$ is approximated by linear interpolation using $\{x_i^\mu, \hat{\boldsymbol{\mu}}(x_i^\mu)\}_{i=1}^{n_\mu}$. For ease of notation, we use $\hat{\boldsymbol{\mu}}$ to denote this interpolated piecewise linear curve later on. Examples can be found in Figure 2 where all the mean curves (black solid) were obtained using the above method.

Estimating an uncertainty region including data points with $\eta$ probability is much more challenging. We addressed this problem by the following heuristic algorithm.

Step 1. Draw $x_i^*$'s from Unif(0,1) independently for $i = 1, \ldots, n_1$;

Step 2. Sample the corresponding $\boldsymbol{y}_i^*$ from the posterior predictive distribution conditional on these latent coordinates $p(\boldsymbol{y}_1^*, \ldots, \boldsymbol{y}_{n_1}^*|x_{1:n_1}^*, \hat{\boldsymbol{x}}, \boldsymbol{y}_{1:n}, \hat{\Theta})$;

Step 3. Repeat steps 1-2 $n_2$ times, collecting all $n_1 \times n_2$ samples $\boldsymbol{y}^*$'s;

Step 4. Find the shortest distances from these $\boldsymbol{y}^*$'s to the posterior mean curve $\hat{\boldsymbol{\mu}}$, and find the $\eta$-quantile of these distances denoted by $\rho$;

Step 5. Moving a radius-$\rho$ ball through the entire curve $\hat{\boldsymbol{\mu}}([0, 1])$, the envelope of the moving trace defines the $\eta\%$ uncertainty band.

Note that step 4 can be easily solved since $\hat{\boldsymbol{\mu}}$ is a piecewise linear curve. Examples can be found in Figure 2, where the 95% uncertainty bands (dotted shading) were found using the above algorithm.

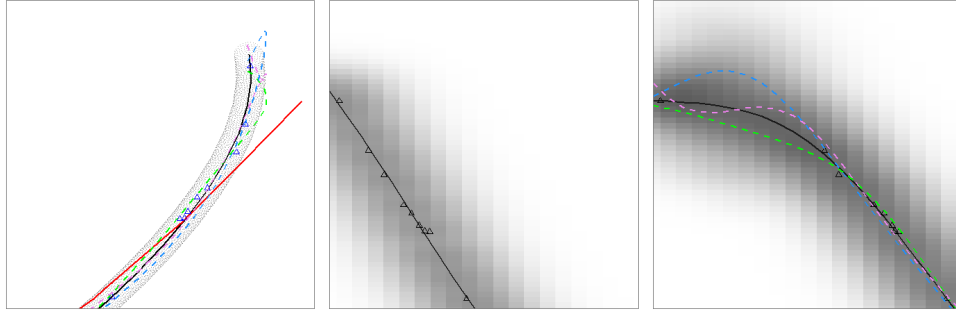

Figure 3: The zoom-in of the spiral case 3 (**left**) and the corresponding coordinate function, $\mu_2(x)$, of electroGP (**middle**) and GP-LVM (**right**). The gray shading denotes the heatmap of the posterior distribution of $(x, y_2)$ and the black curve denotes the posterior mean.

### 3.3 Simulation

In this subsection, we compare the performance of electroGP with GP-LVM and principal curves (P-curve) in simple simulation experiments. 100 data points were sampled from each of the following three 2-dimensional distributions: a Gaussian distribution, a rotated parabola with Gaussian noises and a spiral with Gaussian noises. ElectroGP and GP-LVM were fitted using the same initial values obtained from LLE, and the P-Curve was fitted using the `princurve` package in R.

The performance of the three methods is compared in Figure 2. The dotted shading represents a 95% posterior predictive uncertainty band for a new data point $\boldsymbol{y}_{n+1}$ under the electroGP model. This illustrates that electroGP obtains an excellent fit to the data, provides a good characterization of uncertainty, and accurately captures the concentration near a 1d manifold embedded in two dimensions. The P-curve is plotted in red. The extremely poor representation of P-curve is as expected based on our experience in fitting principal curve in a wide variety of cases; the behavior is highly unstable. In the first two cases, the P-Curve corresponds to a smooth curve through the center of the data, but for the more complex manifold in the third case, the P-Curve is an extremely poor representation. This tendency to cut across large regions of near zero data density for highly curved manifolds is common for P-Curve.

For GP-LVM, we show three random realizations (dashed) from the posterior in each case. It is clear the results are completely unreliable, with the tendency being to place part of the curve through where the data have high density, while also erratically adding extra outside the range of the data. The GP-LVM model does not appropriately penalize such extra parts, and the very poor performance shown in the top right of Figure 2 is not unusual. We find that electroGP in general performs dramatically better than competitors. More simulation results can be found in the supplement. To better illustrate the results for the spiral case 3, we zoom in and present some further comparisons of GP-LVM and electroGP in Figure 3.

As can be seen the right panel, optimizing $x$'s without any constraint results in "holes" on $[0, 1]$. The trajectories of the Gaussian process over these holes will become arbitrary, as illustrated by the three realizations. This arbitrariness will be further projected into the input space $\mathcal{Y}$, resulting in the erratic curve observed in the left panel. Failing to have well spread out $x$'s over $[0, 1]$ not only causes trouble in learning the curve, but also makes the posterior predictive distribution of $\boldsymbol{y}_{n+1}$ overly diffuse near these holes, e.g., the large gray shading area in the right panel. The middle panel shows that electroGP fills in these holes by softly constraining the latent coordinates $x$'s to spread out while still allowing the flexibility of moving them around to find a smooth curve snaking through them.

### 3.4 Prediction

Broad prediction problems can be formulated as the following missing data problem. Assume $m$ new data $\boldsymbol{z}_i$, for $i = 1, \ldots, m$, are partially observed and the missing entries are to be filled in. Letting $\boldsymbol{z}_i^O$ denote the observed data vector and $\boldsymbol{z}_i^M$ denote the missing part, the conditional distribution of

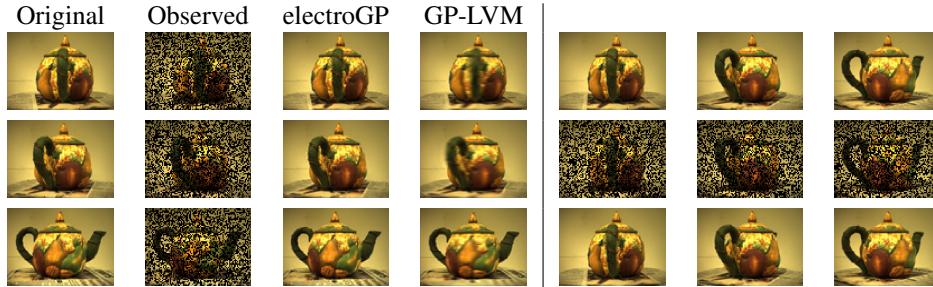

| Original | Observed | electroGP | GP-LVM |
|----------|----------|-----------|--------|

Figure 4: **Left Panel**: Three randomly selected reconstructions using electroGP compared with those using Bayesian GP-LVM; **Right Panel**: Another three reconstructions from electroGP, with the first row presenting the original images, the second row presenting the observed images and the third row presenting the reconstructions.

the missing data is given by

$$p(\boldsymbol{z}_{1:m}^M|\boldsymbol{z}_{1:m}^O, \hat{\boldsymbol{x}}, \boldsymbol{y}_{1:n}, \hat{\Theta})$$
$$= \int_{x_1^z} \cdots \int_{x_m^z} p(\boldsymbol{z}_{1:m}^M|x_{1:m}^z, \hat{\boldsymbol{x}}, \boldsymbol{y}_{1:n}, \hat{\Theta}) \times p(x_{1:m}^z|\boldsymbol{z}_{1:m}^O, \hat{\boldsymbol{x}}, \boldsymbol{y}_{1:n}, \hat{\Theta}) \mathrm{d}x_1^z \cdots \mathrm{d}x_m^z,$$

where $x_i^z$ is the corresponding latent coordinate of $\boldsymbol{z}_i$, for $i = 1, \ldots, n$. However, dealing with $(x_1^z, \ldots, x_m^z)$ jointly is intractable due to the high non-linearity of the Gaussian process, which motivates the following approximation,

$$p(x_{1:m}^z|\boldsymbol{z}_{1:m}^O, \hat{\boldsymbol{x}}, \boldsymbol{y}_{1:n}, \hat{\Theta}) \approx \Pi_{i=1}^m p(x_i^z|\boldsymbol{z}_i^O, \hat{\boldsymbol{x}}, \boldsymbol{y}_{1:n}, \hat{\Theta}).$$

The approximation assumes $(x_1^z, \ldots, x_m^z)$ to be conditionally independent. This assumption is more accurate if $\hat{\boldsymbol{x}}$ is well spread out on $(0, 1)$, as is favored by Corp.

The univariate distribution $p(x_i^z|\boldsymbol{x}_i^O, \boldsymbol{y}_{1:n}, \hat{\boldsymbol{u}}, \hat{\Theta})$, though still intractable, is much easier to deal with. Depending on the purpose of the application, either a Metropolis Hasting algorithm could be adopted to sample from the predictive distribution, or a optimization method could be used to find the MAP of $x^z$'s. The details of both algorithms can be found in the supplement.

## 4   Experiments

**Video-inpainting**    200 consecutive frames (of size $76 \times 101$ with RGB color) [13] were collected from a video of a teapot rotating $180°$. Clearly these images roughly lie on a curve. 190 of the frames were assumed to be fully observed in the natural time order of the video, while the other 10 frames were given without any ordering information. Moreover, half of the pixels of these 10 frames were missing. The electroGP was fitted based on the other 190 frames and was used to reconstruct the broken frames and impute the reconstructed frames into the whole frame series with the correct order. The reconstruction results are presented in Figure 4. As can be seen, the reconstructed images are almost indistinguishable from the original ones. Note that these 10 frames were also correctly imputed into the video with respect to their latent position $x$'s. ElectroGP was compared with Bayesian GP-LVM [7] with the latent dimension set to 1. The reconstruction mean square error (MSE) using electroGP is 70.62, compared to 450.75 using GP-LVM. The comparison is also presented in Figure 4. It can be seen that electroGP outperforms Bayesian GP-LVM in high-resolution precision (e.g., how well they reconstructed the handle of the teapot) since it obtains a much tighter and more precise estimate of the manifold.

**Super-resolution & Denoising**    100 consecutive frames (of size $100 \times 100$ with gray color) were collected from a video of a shrinking shockwave. Frame 51 to 55 were assumed completely missing and the other 95 frames were observed with the original time order with strong white noises. The shockwave is homogeneous in all directions from the center; hence, the frames roughly lie on a curve. The electroGP was applied for two tasks: 1. Frame denoising; 2. Improving resolution by interpolating frames in between the existing frames. Note that the second task is hard since there are

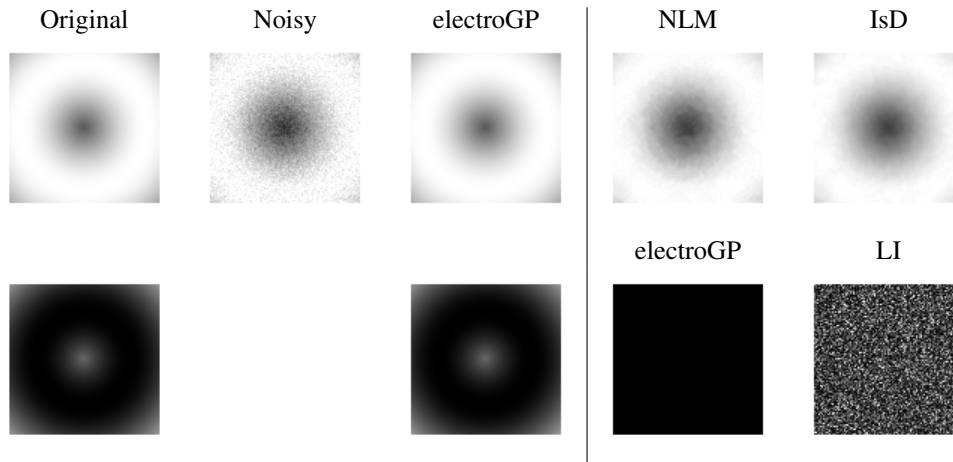

Figure 5: **Row 1**: From left to right are the original 95th frame, its noisy observation, its denoised result by electroGP, NLM and IsD; **Row 2**: From left to right are the original 53th frame, its regeneration by electroGP, the residual image (10 times of the absolute error between the imputation and the original) of electroGP and LI. The blank area denotes its missing observation.

5 consecutive frames missing and they can be interpolated only if the electroGP correctly learns the underlying manifold.

The denoising performance was compared with non-local mean filter (NLM) [14] and isotropic diffusion (IsD) [15]. The interpolation performance was compared with linear interpolation (LI). The comparison is presented in Figure 5. As can be clearly seen, electroGP greatly outperforms other methods since it correctly learned this one-dimensional manifold. To be specific, the denoising MSE using electroGP is only $1.8 \times 10^{-3}$, comparing to 63.37 using NLM and 61.79 using IsD. The MSE of reconstructing the entirely missing frame 53 using electroGP is $2 \times 10^{-5}$ compared to 13 using LI. An online video of the super-resolution result using electroGP can be found in this link[1]. The frame per second (fps) of the generated video under electroGP was tripled compared to the original one. Though over two thirds of the frames are pure generations from electroGP, this new video flows quite smoothly. Another noticeable thing is that the 5 missing frames were perfectly regenerated by electroGP.

## 5   Discussion

Manifold learning has dramatic importance in many applications where high-dimensional data are collected with unknown low dimensional manifold structure. While most of the methods focus on finding lower dimensional summaries or characterizing the joint distribution of the data, there is (to our knowledge) no reliable method for probabilistic learning of the manifold. This turns out to be a daunting problem due to major issues with identifiability leading to unstable and generally poor performance for current probabilistic non-linear dimensionality reduction methods. It is not obvious how to incorporate appropriate geometric constraints to ensure identifiability of the manifold without also enforcing overly-restrictive assumptions about its form.

We tackled this problem in the one-dimensional manifold (curve) case and built a novel electro-static Gaussian process model based on the general framework of GP-LVM by introducing a novel Coulomb repulsive process. Both simulations and real world data experiments showed excellent performance of the proposed model in accurately estimating the manifold while characterizing uncertainty. Indeed, performance gains relative to competitors were dramatic. The proposed electroGP is shown to be applicable to many learning problems including video-inpainting, super-resolution and video-denoising. There are many interesting areas for future study including the development of efficient algorithms for applying the model for multidimensional manifolds, while learning the dimension.

## Footnotes

[1]https://youtu.be/N1BG220J5Js This online video contains no information regarding the authors.

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
