[Supplementary Material]

# Supplementary Material

## Probabilistic Curve Learning: Coulomb Repulsion and the Electrostatic Gaussian Process

**Ye Wang**
Department of Statistics
Duke University
Durham, NC, USA, 27705
eric.ye.wang@duke.edu

**David Dunson**
Department of Statistics
Duke University
Durham, NC, USA, 27705
dunson@stat.duke.edu

## Proof of Lemma 1 and Lemma 2

In this section we let $X_t$, $t \in \mathbb{N}_+$ denote a realization from Corp.

**Lemma 1.** *For any $n \in \mathbb{N}_+$, any $1 \leqslant i < n$ and any $\epsilon > 0$, we have*

$$p(X_n \in \mathcal{B}(X_i, \epsilon)|X_1, \ldots, X_{n-1}) < \frac{2\pi^2 \epsilon^{2r+1}}{2r+1}$$

*where $\mathcal{B}(X_i, \epsilon) = \{X \in (0,1) : d(X, X_i) < \epsilon\}$.*

*Proof.* By definition of the Corp, for $\epsilon$ small enough, we have

$$
\begin{aligned}
&p(X_n \in \mathcal{B}(X_i, \epsilon)|X_1, \ldots, X_{n-1}) \\
=&C \int_{\sin\left(X_n - X_i\right) < \epsilon} \Pi_{j=1}^{n-1} \sin^{2r}\left(\pi X_n - \pi X_j\right) \mathrm{d}X_n \\
\approx&C \int_{|X_n - X_i| < \epsilon} \Pi_{j=1}^{n-1} \sin^{2r}\left(\pi X_n - \pi X_j\right) \mathrm{d}X_n,
\end{aligned}
$$

where $C$ is the normalizing constant. When $X_i \in (\epsilon, 1 - \epsilon)$, the following is true,

$$
\begin{aligned}
&\int_{X_i - \epsilon}^{X_i + \epsilon} \Pi_{j=1}^{n-1} \sin^{2r}\left(\pi X_n - \pi X_j\right) \mathrm{d}X_n \\
\leqslant& \int_{X_i - \epsilon}^{X_i + \epsilon} \sin^{2r}\left(\pi X_n - \pi X_i\right) \mathrm{d}X_n \\
=&2 \int_0^\epsilon \sin^{2r}(\pi x) \mathrm{d}x \\
<&2 \int_0^\epsilon x^{2r} \mathrm{d}x \\
=&\frac{2\pi^2 \epsilon^{2r+1}}{2r+1}.
\end{aligned}
$$

When $X_i \in (0, \epsilon)$, the following is true,

$$\left( \int_0^{X_i+\epsilon} + \int_{1-\epsilon+X_i}^1 \right) \Pi_{j=1}^{n-1} \sin^{2r} \left( \pi X_n - \pi X_j \right) \mathrm{d}X_n$$

$$\leqslant \int_0^{X_i+\epsilon} \sin^{2r} \left( \pi X_n - \pi X_i \right) \mathrm{d}X_n + \int_{1-\epsilon+X_i}^1 \sin^2 \left( \pi - \pi X_n + \pi X_i \right) \mathrm{d}X_n$$

$$= \left( \int_0^\epsilon + \int_0^{X_i} \right) \sin^{2r} (\pi x) \mathrm{d}x + \int_{X_i}^\epsilon \sin^{2r} (\pi x) \mathrm{d}x$$

$$< \frac{2\pi^2 \epsilon^{2r+1}}{2r+1}$$

The proof for $X_i \in (1 - \epsilon, 1)$ is the same as above and hence is neglected here. $\qquad \square$

**Lemma 2.** *For any $n \in \mathbb{N}_+$, $p(X_{t_1}, \ldots, X_{t_k})$ (due to the exchangeability, we can assume $X_1 < X_2 < \cdots < X_n$ without loss of generality) is maximized when and only when*

$$d(X_i, X_{i-1}) = \sin \left( \frac{1}{n+1} \right) \text{ for all } 2 \leqslant i \leqslant n. \tag{1}$$

*Proof.* The log of the density is given up to a constant by

$$l(X_1, \ldots, X_n) \propto \sum_{i>j} c \log \left[ \sin^2 \left( \pi X_i - \pi X_j \right) \right].$$

The first order derivatives are given by

$$\frac{\partial l}{\partial X_i} = \sum_{j \neq i}^n \frac{2c\pi \sin \left( \pi X_i - \pi X_j \right) \cos \left( \pi X_i - \pi X_j \right)}{\sin^2 \left( \pi X_i - \pi X_j \right)}$$

$$= \sum_{j \neq i}^n 2c\pi \cot \left( \pi X_i - \pi X_j \right) \tag{2}$$

For any $X_1 < X_2 < \cdots < X_n$ satisfying condition $d(X_i, X_{i-1}) = \sin \left( \frac{1}{n+1} \right)$, (2) can be rewritten as

$$\sum_{j \neq i}^n 2c\pi \cot \left( \pi X_i - \pi X_j \right)$$

$$= \sum_{j \neq i}^n 2c\pi \cot \left( \frac{i-j}{n} \pi \right)$$

$$= \sum_{j=1}^{i-1} 2c\pi \cot \left( \frac{j}{n} \pi \right) + \sum_{j=i+1}^n 2c\pi \cot \left( -\frac{j-i}{n} \pi \right)$$

$$= \sum_{j=1}^{i-1} 2c\pi \cot \left( \frac{j}{n} \pi \right) + \sum_{j=i+1}^n 2c\pi \cot \left( \frac{n-j+i}{n} \pi \right)$$

$$= \sum_{j=1}^{n-1} 2c\pi \cot \left( \frac{j}{n} \pi \right)$$

$$= 0$$

Hence (1) satisfies the first order condition. The second order derivatives are given by

$$\frac{\partial^2 l}{\partial X_i \partial X_j} = 2c\pi \frac{\partial \left[ \cot \left( \pi X_i - \pi X_j \right) \right]}{\partial X_j}$$

for $i \neq j$ and

$$\frac{\partial^2 l}{\partial X_i^2} = \sum_{j \neq i}^{n} 2c\pi \frac{\partial \left[ \cot \left( \pi X_i - \pi X_j \right) \right]}{\partial X_j}$$

$$= \sum_{j \neq i}^{n} \frac{\partial^2 l}{\partial X_i \partial X_j}$$

Hence the Hessian matrix is positive semi-definite, indicating that (1) is a global maxima. Note also that the Hessian matrix is rank-deficit, indicating that the solution to this maximization problem is not unique. □

## Sampling from Corp

The sampling method can be easily summarized as,

Step 1  Sample $X_1$ from Unif(0,1);

Step 2  Repeatedly sample $X_i$ from $p(X_i|X_1, \ldots, X_{i-1})$ until desired sample size reached.

The difficulty arises in step 2 since

$$p(X_i|X_1, \ldots, X_{i-1}) \propto \Pi_{j=1}^{i-1} \sin^{2r} \left( \pi X_i - \pi X_j \right) \mathbb{1}_{X_i \in (0,1)}$$

is multi-modal and not analytically integrable. Fortunately, sampling from the above univariate distribution can be done by rejection sampling. The only trick here is to find a proper proposal distribution. Naïvely using a uniform would result in very high rejection rate as $i$ grows larger.

Assuming without loss of generosity that $X_1 < X_2 < \ldots < X_{i-1}$, it can be easily checked that there is one local mode within each interval of $(X_j, X_{j+1})$, for $1 \leqslant j \leqslant i-2$. We denote the mode by $p_j$ and the interval by $S_j$. There is also one mode on $(0, X_1) \bigcup (X_{i-1}, 1)$. We denote this mode by $p_{i-1}$, and this interval by $S_{i-1}$. Sampling from this conditional distribution can be summarized as,

Step 1  Sample $k$ from Multinomial$_i(\boldsymbol{a})$ where $a_j = \int_{S_j} p(X_i|X_1, \ldots, X_{i-1}) \mathrm{d}X_i$ for $j = 1, \ldots, i-1$. These integration is done using numerical method;

Step 2  Use Unif$(S_k)$ as the proposal distribution and calculate $p_k$ using numerical maximization method. Use rejection sampling to sample $X_i$ from the truncated conditional distribution $p_{S_k}(X_i|X_1, \ldots, X_{i-1})$.

## Prediction

Assume $m$ new data $\boldsymbol{z}_i$, for $i = 1, \ldots, m$, are partially observed and the missing entries are to be predicted. Letting $\boldsymbol{z}_i^O$ denote the observed data vector and $\boldsymbol{z}_i^M$ denote the missing part. We approximate the predictive distribution by assuming that these $\boldsymbol{z}_i$'s are conditionally independent. For ease of notation, we focus on discussing the prediction algorithm for one partially observed new data vector $(\boldsymbol{z}^O, \boldsymbol{z}^M)$.

**Sample from the posterior predictive distribution**    Instead of sampling from $p(\boldsymbol{z}^M|\boldsymbol{z}^O, \hat{\boldsymbol{x}}, \boldsymbol{y}_{1:n}, \hat{\Theta})$, we sample from $p(\boldsymbol{z}^M, x^z|\boldsymbol{z}^O, \hat{\boldsymbol{x}}, \boldsymbol{y}_{1:n}, \hat{\Theta})$, which can be factorized into two parts $p(\boldsymbol{z}^M|x^z, \boldsymbol{z}^O, \hat{\boldsymbol{x}}, \boldsymbol{y}_{1:n}, \hat{\Theta})$ and $p(x^z|\boldsymbol{z}^O, \hat{\boldsymbol{x}}, \boldsymbol{y}_{1:n}, \hat{\Theta})$. The first part is simply a conditional Gaussian distribution and can be easily sampled. We use the Metropolis Hasting algorithm to sample from the intractable second part, using Unif(0,1) as the proposal distribution. Note that Unif(0,1) is a natural choice, since it is the prior distribution of $x$. It can be easily generalized to a piecewise uniform distribution, as what we did in sampling Corp, to decrease the rejection rate.

**Find the MAP**    MCMC can be infeasible in some applications due to its expensive computation. A straightforward solution is to use EM algorithm treating $x^z$ as an augmented variable, which will give us a point estimate of $\boldsymbol{z}^M$. We propose another heuristic algorithm that would give us instead of point estimate a distribution of $\boldsymbol{z}^M$. The algorithm is very simple and is summarized as follows,

Step 1. Find $\hat{x}^z$ by maximizing $p(x^z|\boldsymbol{z}^O, \hat{\boldsymbol{x}}, \boldsymbol{y}_{1:n}, \hat{\Theta})$;

Step 2. Return $p(\boldsymbol{z}^M|\hat{x}^z, \boldsymbol{z}^O, \hat{\boldsymbol{x}}, \boldsymbol{y}_{1:n}, \hat{\Theta})$, which is simply a multivariate Gaussian.

## Simulation Results

Figure 1: Visualization of two simulation experiments where the data (triangles) are simulated from a rotated sine curve with Gaussian noises (**top**), an arc with Gaussian noises (**bottom**). The dotted shading denotes the 95% posterior predictive uncertainty band of $(y_1, y_2)$ under electroGP. The black curve denotes the posterior mean curve under electroGP and the red curve denotes the P-curve.