[Reviews · NeurIPS 2015]

Submitted by Assigned_Reviewer_1

The paper is interesting and the authors do a good job explaining the idea and the material. Your models seem to be most closely related to the one in [11], where you instead have used the Coulomb repulsive process for the manifold points. Nevertheless, [11] is not included in your overview of related work in the introduction, only its prior work in [6]. Could you more carefully explain how you work relates to [11]?

In the experimental section your one-dimensional manifold corresponds to time, since the video frames constitute a time series. This opens up to comparisons with dynamical models and especially hidden Markov models to describe such data. Could you relate your model to HMMs? By using the sequential order of the video frames, the prediction performance should be increased. Such models would also automatically solve the "gap" issue that you encounter. Further, in the experimental section you state on row 310/311 that "As can be seen the right panel, optimizing x's without any constraint results in "holes" on [0, 1]". To me this is not clear. Please clarify how to see this from the figures. Do also give a clearer fig reference. In addition, you alternate between comparing with GP-LVM and Bayesian GP-LVM in the results sections. Please be consistent or give a motivation for your choices. Finally, are there any situations that the proposed model does not handle? It would be fair to give such comments as well to increase the objectivity of the paper.

I think the model is interesting and it seems to solve problems present in previous related models. The results are promising and worth to be published.

Minors: - Fig 3. the caption should be improved since it does not fully explain the content in the figures. It would be helpful to mark the regions in figure 2 where you have zoomed in. - In fig 4, I think you should exchange rows and columns in the right panel. Now it is a bit confusing and not consistent with the left panel.
Summary: The model is interesting and it seems to solve problems present in previous related models. The results are promising and worth to be published.

Submitted by Assigned_Reviewer_2

=Summary=

This paper proposes a maximum penalized likelihood version of the GP-LVM (equation after eq. 4). The penalty term added to the GP-LVM log likelihood is the log joint probability density function of the inputs under a Coulomb repulsive process.

=Evaluation=

Quality: medium because of the missing references. Clarity: good Originality: good, to the best of my knowledge Significance: medium (early days, still far from an easy to reimplement approach)

=Details=

The optimization depends critically on initialization, and the authors propose a heuristic that relies on using a similarity preserving traditional embedding, as well as a way to initialize the GPs hyperparameters.

Obtaining posterior uncertainty is pretty tedious and there isn't a good solution. The authors propose a heuristic.

Under the GP-LVM model, two latent points that are nearby must be associated with nearby data points. However, there is no guarantee that two nearby data points will be mapped to two latent points that are near one another. This causes many of the ailments of the GP-LVM mentioned in the current paper. The authors should consider citing the constrained likelihood approach, that imposes that the inputs be a smooth map from data to latent [1, 3]. These formulations do not correspond to a proper Bayesian generative model.

There exists however a Bayesian formulation of the GP-LVM [2] the authors should relate their method to.

Minor:

- figure 2 is really hard to read.

Refs:

[1] "Local Distance Preservation in the GP-LVM Through Back Constraints" Neil Lawrence and Joaquin Quinonero Candela Proceedings of the 23rd International Conference on Machine Learning 2006, pages 512--520, Pittsburgh, PA

[2]

"Bayesian Gaussian process latent variable model"

M. K. Titsias and N. D. Lawrence. (2010) in Y. W. Teh and D. M. Titterington (eds) Proceedings of the Thirteenth International Workshop on Artificial Intelligence and Statistics, JMLR W&CP 9, Chia Laguna Resort, Sardinia, Italy, pp 844--851.

[3] "Topologically-constrained latent variable models"

R. Urtasun, D. J. Fleet, A. Geiger, J. Popovi, T. J. Darrell and N. D. Lawrence. (2008) in S. Roweis and A. Mccallum (eds) Proceedings of the International Conference in Machine Learning, Omnipress, , pp 1080-1087.
Summary: An interesting penalized maximum likelihood version of the GP-LVM to force better distributed embeddings. The paper suffers from not having referenced more modern versions of the GP-LVM which in different ways attempt to improve the properties of the learnt embeddings, and to a smaller extent from a somewhat tedious heuristic for computing posterior embedding uncertainty.

Submitted by Assigned_Reviewer_3

This paper proposes a new method to learn a mapping function from a d-dimensional data point, y, to one-dimensional embedding, x.

The paper extends the standard GP to a hierarchical model by incorporating what they call Coulomb repulsive process (Corp). It first generates samples of the one-dimensional x and then generates observable d-dim data points through a GP, whose covariance is constrained to be diagonal.

Intuitively, Corp is used to ensure reasonable inter-point distance in the embedding space through the repulsive force in [0,1] to make the distance of 0.5 is the least stressful state. The equi-distance prior in [0,1] looks like a natural choice when the original distribution for y forms a compact single continuous manifold in the original space.

To find the embeddings, they propose an alternating algorithm, where GP parameters and the embeddings are solved alternatingly.

To the best of my knowledge, the use of Corp in the context of GP-LVM is new. The proposed approach looks interesting enough to accept the paper.

I personally think that the paper can be improved by describing theoretical aspects of the model more carefully. It looks obvious to me that the model still has major limitations such as the strong dependency on the initial guess (x_0). I feel that the authors are overselling the model by making too strong statements on the practical utility of the model.

For example, in practice, we all know that careful pre-process is required to make "your favorite distance-preserving manifold" work. What if the distribution looks multi-modal? What if x_0 misses to capture the major features of the original distribution? Your claim in Sec. 5 is scientifically sound?

I know recently more and more reviewers tend to demand extensive experimental comparison. However, for papers like this, the most important thing is to build a common understanding with the audience on the major features of the model and wide applicability of the model. I hope the camera-ready version gets more understandable and sophisticated by addressing it.

Minor comments. - Do not mix between bold and non-bold notation to represent vectors. Use boldface for e.g. \mu in Introduction.

- Introduction of Corp in Sec.2 can be improved. This is basically to introduce the repulsive force. Don't make it overly mathematical.

- If you are to use r=1, you don't have to show r=20. You can show illustrative figures instead.

- Dependency on x_0 should be discussed in depth. If you cannot, do not overclaim the practical utility.

- Also, describe practical limitations in addition to practical advantages.

- Sec. 5 sounds like a frivolous sales talk. This is a good paper as is. You don't have to worry about the "more experiment!" criticism. I'd like you to make more sophisticated academic comments.

- Redraw Fig.2. Visibility is extremely bad.
Summary: The paper proposes a new prior to GP-LVM to address well-known shortcomings of GP-LVM to realize a natural distribution of embedded coordinates. The paper seems to be rather overclaiming the practical utility of the proposed method,

but the approach looks novel enough to be accepted.

Author Feedback
Author rebuttal: We thank the reviewers for the time and expertise they have invested in providing their feedback.

To Reviewer 1:
We chose [6] as a direct comparison with electroGP since they both try to find the MAP of the latent x's. We agree that the relation to [11] should be discussed and will include it in the introduction of the final version.

By "hole" we meant, in the right panel of Figure 3, the vacant space between the leftmost and the second leftmost x's (triangle).

The proposed model is currently constrained to the simplest scenario, which is to learn a compact smooth curve. We hope to explore its ability of handling more complex situation in the future. We also want to thank the reviewer for the comments on the clearness of the figures and will improve it in our final version.

To Reviewer 2:
We agree with the reviewer and will explore and include the mentioned references in the final version. And we will redraw Figure 2.

We also agree that the optimization relies heavily on initial ordering of x's. Fortunately, in our experience, the traditional manifold learning methods provide good enough initialization. This is probably due to that we are only solving the simplest one-dimensional curve learning task. We admit that the model is still in its early days. Generalizing it to multi-dimensional manifold problems and relaxing the dependency on initialization are both on our to-do list.

To Reviewer 3:
We thank the reviewer for pointing out the limitations of our paper. Exploring the theoretical properties of the model and relaxing the dependency on the initialization are both on our to-do list.

The reviewer also raised a good point of the potential multi-modality in the data, which is not concerned in this paper since we only consider a simple scenario where the data live on a smooth compact curve. This is certainly a limitation and might be solved by a mixture of electroGP. We hope to explore this possibility in the future.
We also thank the reviewer for the helpful comments and will try our best to address them in the final version.

To Reviewer 4, 6 and 7:
We thank the reviewers for their accurate summaries of this paper.